# Incremental3D: Real-time Incremental 3D Scene Generation with Scene Graphs

**Penggang Gao**                                                  *penggang.gao@iit.it*
*Advanced Robotics*
*Istituto Italiano di Tecnologia (IIT) & Universitá di Genova, Italy*

**Yonas Teodros Tefera**                                          *yonas.tefera@iit.it*
*Advanced Robotics*
*Istituto Italiano di Tecnologia (IIT), Italy*

**Darwin G. Caldwell**                                            *darwin.caldwell@iit.it*
*Advanced Robotics*
*Istituto Italiano di Tecnologia (IIT), Italy*

**Nikhil Deshpande**                                             *nikhil.deshpande@nottingham.ac.uk*
*School of Computer Science*
*University of Nottingham, UK*

**Reviewed on OpenReview:** *https://openreview.net/forum?id=am8Zv3R8GW*

## Abstract

Realistic 3D environments are important for a wide range of applications, including robotics, simulation, virtual reality, and video games. The goal of 3D scene generation is to create spatially structured, semantically meaningful, and visually realistic environments that capture objects and their relationships in space. Graph-based 3D scene generation approaches represent environments as scene graphs, where nodes correspond to objects and edges encode their semantic and spatial relationships. However, existing methods become inefficient when the 3D scene graph evolves incrementally, because they are fundamentally single-shot: inserting even a single new object requires regenerating the entire scene. This global re-computation incurs prohibitive latency and scalability limitations. To address this limitation, we propose *Incremental3D*, a framework for incremental 3D scene generation in real time from evolving scene graphs. *Incremental3D* augments the scene graph with a global context node that captures a holistic representation of the evolving environment. At each update step, this node aggregates information from new nodes and edges to form a global embedding. Newly inserted objects are then generated by conditioning on both this embedding and their local features, enabling geometry synthesis and spatial prediction without recomputing unchanged regions. Extensive experiments demonstrate that *Incremental3D* achieves a generation rate of 38 Hz, while maintaining high spatial and geometric accuracy, indicating its potential for real-time and latency-sensitive applications.

## 1 Introduction

The aim of graph-based 3D scene generation (Dhamo et al., 2021; Zhou et al., 2019; Zhai et al., 2023; 2024; Yang et al., 2025; Chattopadhyay et al., 2023) is to synthesize 3D scenes conditioned on scene graphs, where nodes encode object attributes (e.g., position, orientation, and size) and edges represent explicit inter-object relationships (e.g., same-material-as, left-of, larger-than). This paradigm has been widely applied in domains such as game development, virtual reality, and interior design (Dhamo et al., 2021; Samuelson et al., 2025; Yang et al., 2025).

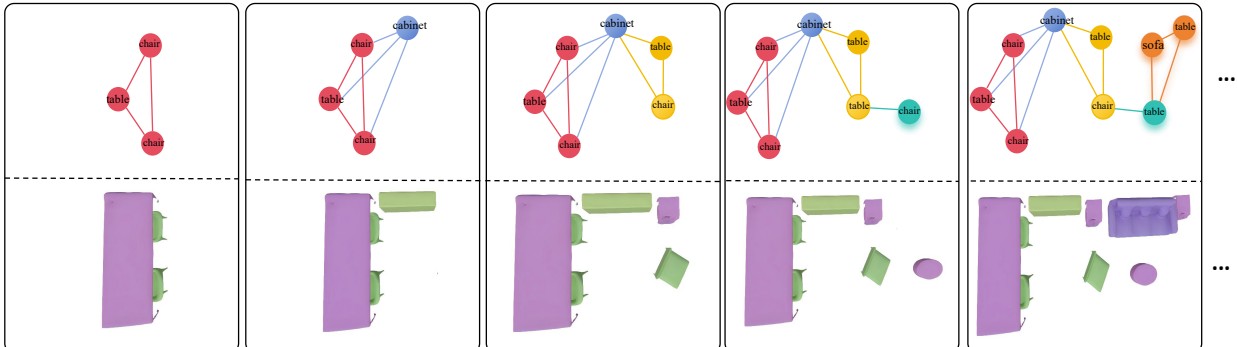

Figure 1: Example of 3D scene generation from an incremental scene graph. When a new object is inserted, the proposed method locally updates the scene by generating the corresponding object meshes while maintaining their spatial and geometric accuracy.

Recent advances in simultaneous localization and mapping (SLAM) (Hughes et al., 2022; Maggio et al., 2024), and 3D scene-graph learning (Wu et al., 2021; Feng et al., 2025; Renz et al., 2025) make it feasible to obtain 3D scene graphs directly from RGB-D or point-cloud inputs, being updated in an incremental manner, as illustrated in Figure 1 (see Appendix A.1 for examples of incremental scene graph construction). However, generating 3D scenes from such incremental scene graph sequences remains an underexplored problem. Existing graph-based 3D scene generation approaches are fundamentally single-shot (Dhamo et al., 2021; Zhou et al., 2019; Zhai et al., 2023; 2024; Yang et al., 2025; Chattopadhyay et al., 2023): inserting even a single new object requires regenerating the entire scene, resulting in substantial computational overhead, causing latency and scalability issues. Even approaches that support scene manipulation still reconstruct the full scene after each node addition. For example, CommonScenes (Zhai et al., 2023) requires over 20 seconds to generate a single room-scale scene. Moreover, these generative methods (Dhamo et al., 2021; Zhou et al., 2019; Zhai et al., 2023; 2024; Yang et al., 2025; Chattopadhyay et al., 2023) synthesize diverse scenes by sampling from learned distributions rather than reconstructing the original environment structure, which poses a challenge to the consistency of the generated scene.

To address this gap, we introduce *Incremental3D*, a framework for incremental 3D scene generation from evolving scene graphs in real time. The key idea is a *global context embedding*, inspired by the CLS (short for *classification*) token used in BERT (Devlin et al., 2019) and Vision Transformer (ViT) (Dosovitskiy et al., 2020). We adapt this mechanism to incremental scene graphs by introducing a global context node, referred to as the *CLS node*, which aggregates holistic scene information and propagates it to newly added nodes. When a new object is inserted into the scene, *Incremental3D* conditions generation on both local object features and the global scene embedding, enabling geometry synthesis and spatial placement without recomputing unchanged regions of the scene. A second design choice focuses on improving geometric and spatial accuracy. Instead of sampling-based generative models such as VAEs or diffusion models (Zhai et al., 2023; 2024; Yang et al., 2025), which emphasize diversity, we adopt an autoencoder-based representation (Li et al., 2023) to better reconstruct the original environment structure.

To enable training and evaluation of *Incremental3D* scene generation, we construct incremental scene graph sequences, based on synthetic data derived from SG-FRONT (Zhai et al., 2024). Our pipeline generates incremental updates through three stages: (1) clustering objects based on Euclidean proximity, (2) constructing a cluster graph to preserve spatial adjacency, and (3) determining an incremental insertion order using a Hamiltonian-path approximation. To further improve robustness to long sequences, we adopt a curriculum learning strategy that gradually increases sequence length and difficulty during training. We evaluate our method against three graph-based 3D scene generation baselines. Our approach achieves a generation rate of 38 Hz while maintaining high layout and geometric accuracy. These results demonstrate the potential of incremental graph-to-3D scene generation for latency-sensitive applications, e.g., teleoperation (Naceri et al., 2021).

Our contributions are summarized as follows:

1. *Incremental3D*: a real-time framework capable of generating 3D scenes from incremental graph updates, achieving 38 Hz generation while preserving high spatial and geometric accuracy.

2. Introducing a curriculum-learning strategy that progressively increases scene complexity, improving accuracy on long-horizon incremental sequences.

3. Extensive experiments showing that *Incremental3D* consistently outperforms single-shot baselines in layout accuracy, geometric accuracy, and real-time performance.

## 2 Related Work

***A. 3D Incremental Scene Graphs:*** Representing 3D environments requires a structured way to capture both the objects present and the relationships that organize them. One effective strategy is to use graph-based abstractions, which model spatial and semantic interactions in a unified framework. Earlier methods in this domain predominantly focused on reconstructing a full scene graph from complete environment data, relying on point clouds (Wald et al., 2020), 3D meshes (Armeni et al., 2019), or image collections (Gay et al., 2019). More recently, researchers have moved to *incremental* graph construction, in which the graph is updated continuously as new frames arrive. For instance, Wu et al. (2023) combines geometric segmentation with graph learning to build scene graphs online from RGB-D streams. This paradigm has been extended to large-scale environments: from room- to building-level (Hughes et al., 2022; Maggio et al., 2024), and even to large-scale outdoor environments (Deng et al., 2024; Samuelson et al., 2025).

***B. 3D Scene Generation from Scene Graphs:*** Building on their strong representational capabilities, scene graphs have become a central intermediate structure for generating 3D environments. Their compact yet expressive formulation allows generative models to condition on both object-level attributes and relational constraints, enabling the synthesis of coherent, semantically grounded scenes. The Graph-to-3D (Dhamo et al., 2021) is the first method to generate 3D geometry directly from a scene graph in an end-to-end way, decoding predicted shape codes with DeepSDF (Park et al., 2019). CommonScenes (Zhai et al., 2023) replaces the codebook with a latent diffusion model, while EchoScene (Zhai et al., 2024) introduces an "information-echo" mechanism that lets nodes information-share diffusion progress. MMGDreamer (Yang et al., 2025) augments text-based graphs with visual cues to improve fidelity. These methods, aimed mainly at architectural or interior design, rely on VAE (Kingma et al., 2013; Kusner et al., 2017) or diffusion priors (Sohl-Dickstein et al., 2015). Random latent sampling in these methods can yield diverse object appearances, but neglects the spatial and geometric accuracy of the scene. In addition, they usually regenerate the entire scene even for a single new node insertion. By contrast, our method aims to provide real-time, purely local generation for 3D objects, incrementally, with high accuracy.

***C. Global Context Embedding:*** Understanding or generating complex structures often requires more than just local information; local features alone can be ambiguous or inconsistent without knowledge of how they fit into the broader whole. Capturing information that extends beyond local neighbourhoods is therefore essential for maintaining semantic consistency, enforcing long-range dependencies, and ensuring that local predictions align with the global structure, whether in language, vision, or 3D environments. Early approaches often relied on simple descriptive statistics to obtain a global summary. For example, max pooling (Reimers & Gurevych, 2019) retains only the most dominant local feature, potentially discarding important contextual cues, while mean pooling assigns equal weight to all elements, failing to reflect the fact that nearby or semantically relevant objects typically exert a stronger influence on newly generated content. More advanced methods replaced these fixed aggregation rules with learnable mechanisms. Attention pooling (Zhang et al., 2021), for instance, introduces a set of learnable query vectors that attend to the entire sequence via multi-head attention, selectively aggregating salient information. However, such approaches require storing all previously generated features and attending to them all, which limits scalability in incremental settings. A widely adopted alternative is the CLS token, used in BERT (Devlin et al., 2019). During self-attention, the CLS token interacts with every other token, absorbing contextual information that is later used for classification. Vision Transformer (ViT) (Dosovitskiy et al., 2020) employs the same device, treating the

CLS embedding as a global image descriptor. This mechanism is now standard for capturing global context in Transformers (Wang et al., 2024), powering tasks such as sentiment analysis, text categorization, and intent recognition. In our proposed framework, the CLS embedding is used to represent the entire 3D scene; newly inserted objects are generated conditionally on this global vector, enabling efficient local updates.

***D. Curriculum Learning:*** Learning complex tasks directly from difficult examples can overwhelm a model, lead to unstable optimization, or hinder its ability to generalize. Curriculum learning (CL) (Bengio et al., 2009) addresses this challenge by structuring the training process in a way that mirrors human learning: starting with simpler, more easily learnable examples and gradually progressing toward harder ones. In domains such as computer vision and natural language processing, CL has proven highly effective in improving model generalization, training stability, convergence speed, and final accuracy (Wang et al., 2021).

## 3  Methodology

*Incremental3D* proposes a framework for real-time 3D scene generation from incrementally evolving scene graphs. Unlike conventional single-shot approaches that regenerate the entire scene after each update, *Incremental3D* synthesizes only the newly inserted objects while preserving the previously generated scene content.

We formalize an incremental scene graph as

$$G = (O_{\text{old}}, R_{\text{old}}; O_{\text{new}}, R_{\text{new}}) \tag{1}$$

, where $O_{\text{old}}$ and $R_{\text{old}}$ denote the objects and relationships accumulated from previous timesteps, $O_{\text{new}}$ and $R_{\text{new}}$ represent the newly added ones. At each update step, the goal is to generate the corresponding 3D objects for $O_{\text{new}}$ while maintaining consistency with the existing scene structure.

As illustrated in Figure 2, our framework follows an autoencoder architecture consisting of a graph encoder and a geometry decoder. The input scene graph is augmented with a CLS node, which acts as a global context token that aggregates information from both existing and newly inserted nodes and edges. This node provides a compact global scene embedding that captures the current state of the environment. The encoder (Sec. 3.1) processes the incremental scene graph and produces embeddings for newly added nodes, while simultaneously updating the global scene embedding $E_g$ through the CLS node. The decoder (Sec. 3.2) then combines the embeddings of the new nodes with the global scene embedding to predict, for each inserted object, its geometric parameters—position, size, and orientation $(p, s, a)$ as well as a shape-specific latent code. The predicted latent codes are used to retrieve semantically and geometrically consistent 3D models from a database, which are then placed and scaled according to the predicted geometric parameters. To improve the incremental generation ability of the network in further steps, it is trained by a curriculum learning strategy based on the number of incremental steps (Section 3.3).

### 3.1  3D Scene–Graph Encoder

Given a 3D incremental scene graph $G$, the encoder $E_c$ extracts semantic and geometric representations, producing two outputs: (1) embeddings $\phi_{v_i}$ for the newly inserted nodes, and (2) a global scene representation $E_g$ that summarizes the overall scene context. To model the relationships between objects, the encoder is implemented as a Triplet Graph Convolutional Network (T-GCN), which jointly encodes subject–predicate–object triplets. Each node in the scene graph represents a 3D object and is defined as:

$$n_i = \big(o_i, p_i, s_i, a_i, \text{shape}_i\big) \tag{2}$$

, where $o_i$ denotes the object's semantic class, and $(p_i, s_i, a_i, \text{shape}_i)$ represent its geometric attributes, including position, size, orientation, and a shape descriptor. These node features provide both semantic and spatial information that the encoder uses to compute object embeddings and update the global scene representation.

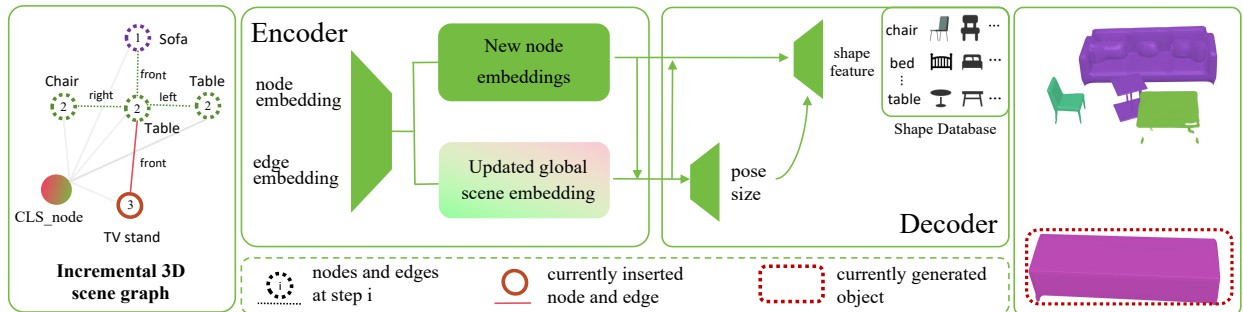

Figure 2: Proposed *Incremental3D* framework. The incremental 3D scene graph is augmented with a CLS node that aggregates information from newly inserted nodes and edges to produce a global scene embedding in the encoder. Conditioned on this global embedding and local object features, new objects are generated incrementally in the decoder via database retrieval.

Inspired by the CLS token in BERT (Devlin et al., 2019) and ViT (Dosovitskiy et al., 2020)—which introduces an extra token attending to all words or patches to capture global context—we add a CLS node to the incremental 3D scene graph and connect it bidirectionally to all newly inserted object nodes. Moreover, old nodes that are directly connected to the new ones are also linked to the CLS node at each step, enabling it to capture information from the newly formed edges. The resulting augmented subgraph is then processed by a graph neural network (GNN). For every directed triplet $(v_i \xrightarrow{e_{i \to j}} v_j)$ of GNN, a layer $l$ performs three operations:

1. Message passing and Edge updates

$$\left(\psi_i^l,\ \phi_{e_{i\to j}}^{l+1},\ \psi_j^l\right) = g_1\left(\phi_{v_i}^l,\ \phi_{e_{i\to j}}^l,\ \phi_{v_j}^l\right), \tag{3}$$

with a shared MLP $g_1$.

2. Neighbor aggregation

$$\rho_i^l = \frac{1}{M_i}\left(\sum_{j\in\mathcal{R}_{\text{out}}} \psi_{\text{out},ij}^l + \sum_{j\in\mathcal{R}_{\text{in}}} \psi_{\text{in},ji}^l\right), \tag{4}$$

where $M_i$ is the total number of incident relations of node $i$. $R_{\text{in}}$, $R_{\text{out}}$ are the set of edges of the node as out(in)-bound objects.

3. Node update

$$\phi_{v_i}^{l+1} = \phi_{v_i}^l + g_2\left(\rho_i^l\right), \tag{5}$$

with another MLP $g_2$.

At each insertion step, the CLS node gathers the information from new nodes and new edges, updating its hidden state to yield a compact, up-to-date scene embedding $E_g$.

### 3.2 3D Scene Decoder

The inputs to the decoder are class label $o_i$, node embedding $\phi_{v_i}$, and global scene embedding $E_g$. The output consists of the predicted bounding box, $(p_i, s_i, a_i)$, and shape code $shape_i$, for the new object. During this stage, bidirectional edges are established between the CLS node and the newly added nodes, enabling the decoder to exploit global scene context for object generation. To preserve the information carried by the new edges, the CLS node is also connected bidirectionally to old nodes that are directly linked to the new nodes at each insertion step. The resulting subgraph is then processed by two consecutive triplet graph convolutional network (GCN) blocks:

- **Box-GCN** ($D_b$): predicts the object position $p$, orientation $a$, and size $s$ from the new node features and the global scene embedding.

- **Shape-GCN** ($D_s$): predicts a shape code conditioned on the layout predicted by Box-GCN $D_b$, the new node features, and the global scene embedding. The resulting new object latent codes are used to query a 3D asset database to retrieve semantically and geometrically consistent models, which are then instantiated in the scene with the predicted poses and sizes.

### 3.3 Training Objectives

At every incremental step, gradients are propagated *only* through the newly inserted objects. For the $N$ new objects, the following loss is minimised:

$$L_{\text{layout}} = \frac{1}{N} \sum_{i=1}^{N} \Big( \|\hat{\mathbf{b}}_i - \mathbf{b}_i\|_1 \ + \ \|\hat{\alpha}_i - \alpha_i\|_1 \Big), \tag{6}$$

$$L_{\text{shape}} = \frac{1}{N} \sum_{i=1}^{N} \|\hat{\mathbf{f}}_i^s - \mathbf{f}_i^s\|_1, \tag{7}$$

$$L_{\text{total}} = \lambda_1 L_{\text{layout}} \ + \ \lambda_2 L_{\text{shape}}. \tag{8}$$

$L_{\text{total}}$ consists of two parts: $L_{\text{layout}}$ and $L_{\text{shape}}$. $\lambda_1$ and $\lambda_2$ are weighting factors. $L_{\text{layout}}$ reconstructs the 3D bounding-box parameters. Here $\mathbf{b}_i \in \mathbb{R}^6$ denotes the box center position and size, $\alpha_i$ the yaw angle, and $N$ the total number of boxes; hats indicate predicted values. $L_{\text{shape}}$ reconstructs the object shape in feature space based on semantic label, object size, and relationship of 3D scene graph. Here, $f_i$ is the feature of the object.

Furthermore, the curriculum learning strategy organizes the training scene samples from easy to hard, where the level of difficulty is defined based on the number of incremental steps across batches.

## 4 Experimental Setup

This section outlines the approach used to implement the evaluation methodology for *Incremental3D*, starting with generating the incremental 3D scene dataset to enable training and testing of the proposed approach.

### 4.1 Incremental Dataset

The incremental 3D scene sequences are constructed using the SG–FRONT dataset (Zhai et al., 2024), which is an extension of the refined 3D–FRONT dataset (Fu et al., 2021) where each room is annotated with a complete 3D scene graph. The dataset covers four scene categories: 4,041 *bedroom*, 813 *living room*, 900 *dining room*, and *all*. The *all* category is composed of a mixed subset of scenes sampled from the other three categories. The training, validation, and test splits follow the same partitioning as the original dataset (Zhai et al., 2024).

The pipeline for incremental dataset creation is illustrated in Figure 3. It consists of three main steps: 1. Given a room, the objects are clustered using DBSCAN (Khan et al., 2014). 2. Each cluster is treated as a node, with edges connecting spatially adjacent clusters to form a graph. 3. The cluster farthest from the global centroid is chosen as the entry point. A visiting order over the remaining clusters is then determined by a nearest-neighbour greedy tour followed by a single 2–opt refinement, producing a near-optimal Hamiltonian path (Sipser, 1996) in $O(N^2)$ time.

Traversing this ordered list yields an incremental sequence of steps and at each step, the current cluster is regarded as the *new-node* set. The edge of the 3D scene-graph, whose subject and object both lie within the same cluster, and the directly connected cluster to it are preserved, while all others are discarded. Our processed relations are sparse, better reflecting real-world incremental scene graph construction (Wu et al.,

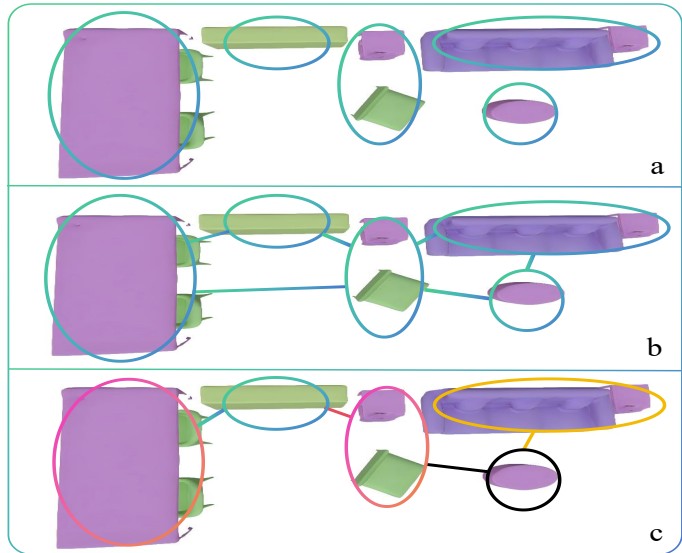

Figure 3: Pipeline of incremental dataset creation. The process includes clustering, graph construction, and insertion order computation, as illustrated in subfigures (a–c).

2021; Hughes et al., 2022; Maggio et al., 2024; Deng et al., 2024), in cases where computational resources are limited and camera fields-of-view are restricted, preventing dense graphs. For subsequent training, the indices of the retained relationships are re-mapped. The resulting sequences cover the entire room without spatial jumps or node revisits.

## 4.2 Compared Baselines

**CommonScenes.** Current graph-based 3D scene generation methods (Dhamo et al., 2021; Zhou et al., 2019; Zhai et al., 2023; 2024; Yang et al., 2025; Chattopadhyay et al., 2023) regenerate the entire scene at each step. Among them, *CommonScenes* (Zhai et al., 2023) is a representative state-of-the-art approach and serves as our primary comparison baseline.

**Baseline 1.** Baseline 1 isolates the effect of *full regeneration*, in which the entire 3D scene is reconstructed from scratch at every incremental step. To ensure a fair comparison, we remove the main source of latency in *CommonScenes*—its online 3D shape generation—and replace it with a shape-retrieval pipeline. Furthermore, while *CommonScenes* employs diffusion- and VAE-based models to encourage scene diversity, Baseline 1 substitutes these components with an autoencoder to produce reconstructions that more closely match the ground truth. Baseline 1 retains the same *full-regeneration* paradigm: whenever new nodes are inserted, *all* existing nodes are re-predicted, including their positions, sizes, orientations, and shape features.

**Baseline 2.** Baseline 2 evaluates the importance of the CLS node and its associated global embedding. At each incremental step, newly inserted nodes generate objects based solely on their *local* neighborhood. In the encoder, new nodes communicate only with directly connected old nodes, without receiving indirect or global contextual information. In the decoder, object generation for new nodes is conditioned solely on their own representations and those of their directly connected neighbors. As in Baseline 1, 3D shapes are obtained via retrieval rather than online generation. This baseline therefore represents a purely local incremental update mechanism that lacks global scene awareness.

## 4.3 Implementation Details

We conduct the training, evaluation, and visualization of *Incremental3D*, baseline 1, and baseline 2 on a single NVIDIA GeForce RTX 3070 GPU with 8 GB memory. We adopt the Adam optimizer with a learning rate of 1e-4 to train the network. $\lambda_1$ and $\lambda_2$ are set as 5 and 1 respectively. *CommonScenes* (Zhai et al.,

2023) is evaluated on an A6000 GPU, since loading CLIP features (Radford et al., 2021) requires substantial memory.

## 4.4 Evaluation Metrics

*Running time* is used as one of the primary evaluation metrics for assessing the efficiency of 3D scene generation, since higher update frequencies enable a broader range of real-time applications. For example, many robotics and visualization tasks require update rates of around 30 Hz. An accurate environment model is essential for enabling robots to perceive or interact effectively with the physical world (Parosi et al., 2023). Therefore, we prioritize accuracy as another key evaluation criterion, covering both layout accuracy and shape accuracy. *Layout accuracy* measures the geometric errors between predicted and ground-truth layouts, including position difference $E_{\mathrm{pos}}$, size difference $E_{\mathrm{size}}$, and angle difference $E_{\mathrm{ang}}$.

$$E_{\mathrm{pos}} = \frac{1}{N} \sum_{i=1}^{N} \left\| p_{\mathrm{pre}}^{(i)} - p_g^{(i)} \right\|_2 \tag{9}$$

$$E_{\mathrm{size}} = \frac{1}{N} \sum_{i=1}^{N} \left\| s_{\mathrm{pre}}^{(i)} - s_g^{(i)} \right\|_2 \tag{10}$$

$$E_{\mathrm{ang}} = \frac{1}{N} \sum_{i=1}^{N} |a_{\mathrm{pre}}^{(i)} - a_g^{(i)}| \tag{11}$$

*Shape accuracy* measures the similarity between the generated object mesh and the ground truth mesh. It is assessed using two metrics that evaluate the quality of the generated objects. Chamfer Distance (CD) (Bakshi et al., 2023) measures the average nearest-neighbor distance between two point clouds and reflects the global geometric alignment, as defined below:

$$\mathrm{CD} = \frac{1}{N} \sum_{i=1}^{N} \min_j \|p_i - g_j\|_2^2 + \frac{1}{N} \sum_{j=1}^{N} \min_i \|g_j - p_i\|_2^2 \tag{12}$$

F-score (Katz et al., 2007) evaluates the degree of local geometric alignment between two meshes by combining precision $P$ and recall $R$, where $d_1$ and $d_2$ denote the nearest distances from predicted points to ground truth points and vice versa—a match is counted if the distance is below a threshold $t$. In our experiments, we set $t = 0.1$ meters.

$$F = \frac{2PR}{P + R}, \tag{13}$$

$$P = \frac{\#\{d_1 < t\}}{N}, \quad R = \frac{\#\{d_2 < t\}}{N} \tag{14}$$

## 5 Results and Analysis

### 5.1 Running time

Table 1 reports the running time (from 3D scene graph input to 3D mesh output) of the baselines and *Incremental3D*. Even with such high-end hardware, *CommonScenes* still takes more than 20 seconds to generate a complete 3D scene from a scene graph, making it impractical for real-time robotic applications. Baseline 1 regenerates all objects at each node insertion, limiting its update rate to 10.02 Hz. Baseline 2 can achieve real-time generation of 3D scenes, but its scene accuracy remains unsatisfactory (see layout and shape accuracy evaluation). *Incremental3D* achieves a generation frequency of 38Hz on average in different scenarios, satisfying the requirements for real-time performance.

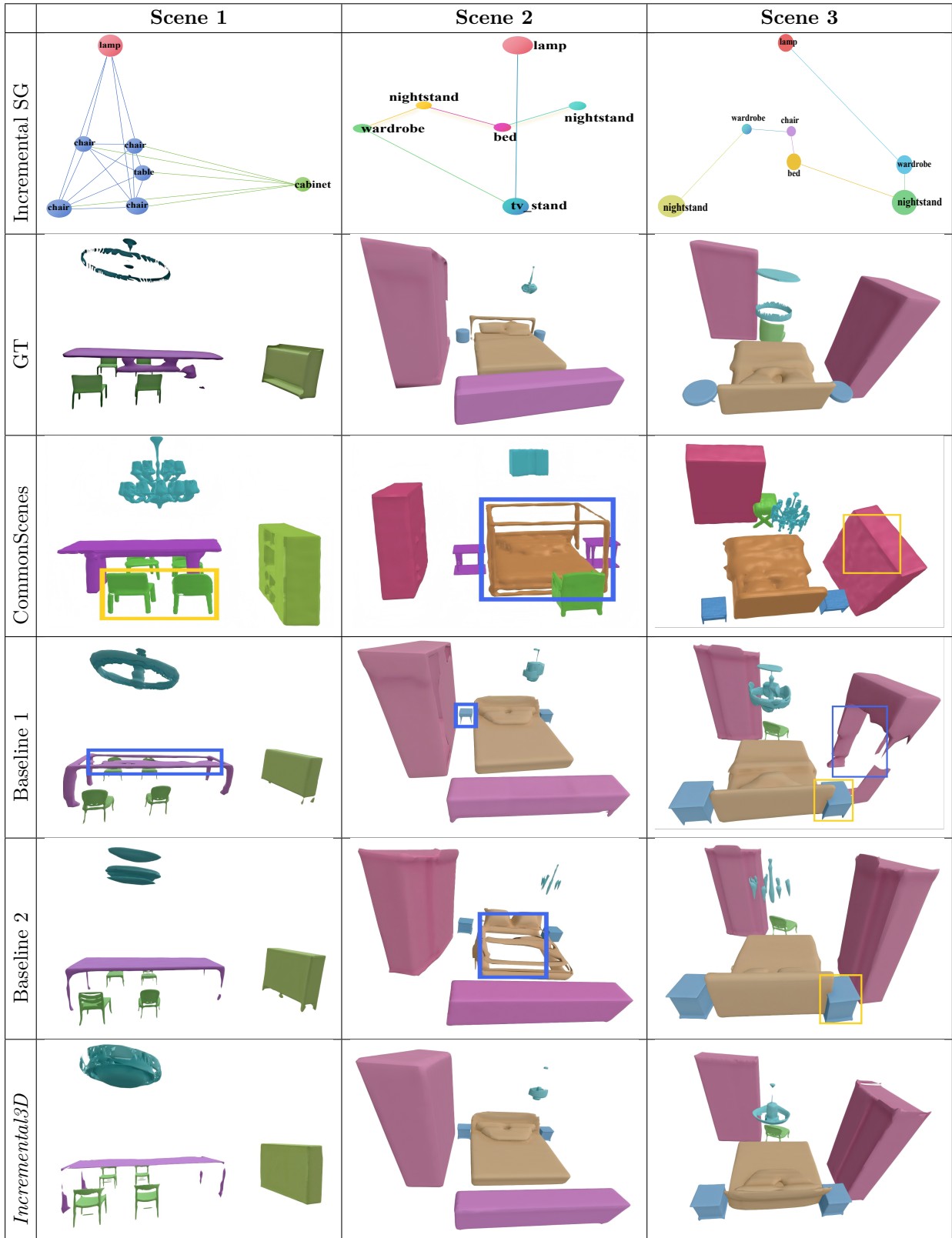

Figure 4: Qualitative comparison across three scenes. Each cell shows one reconstructed 3D scene. Columns correspond to different scenes and rows to different methods. Red boxes highlight layout errors, and blue boxes indicate shape inaccuracies relative to the ground truth.

| Method | Bedroom | Living room | Dining room | All |
|---|---|---|---|---|
| CommonScenes | 24.676 | 21.390 | 24.950 | 28.633 |
| Baseline 1 | 0.079 | 0.114 | 0.116 | 0.090 |
| Baseline 2 | **0.021** | **0.028** | **0.027** | **0.024** |
| *Incremental3D* | 0.022 | 0.029 | 0.028 | 0.026 |

Table 1: Comparison of methods on running time and the unit is seconds.

| Method | Bedroom | | | Living room | | | Dining room | | | All | | |
|---|---|---|---|---|---|---|---|---|---|---|---|---|
| | pos | size | angle | pos | size | angle | pos | size | angle | pos | size | angle |
| CommonScenes | 25.4 | 25.9 | 10.0 | 36.8 | 39.4 | 15.0 | 34.1 | 56.1 | 12.8 | 52.2 | 52.2 | 17.7 |
| Baseline 1 | 4.8 | 4.6 | 3.4 | 15.6 | 9.0 | 4.3 | 14.1 | 7.1 | 7.8 | 3.3 | 3.1 | 2.7 |
| Baseline 2 | 3.9 | 2.9 | 3.3 | 7.9 | 7.6 | 4.9 | 24.0 | 10.7 | 4.4 | 3.9 | 3.7 | 2.9 |
| *Incremental3D* | **1.9** | **1.6** | **1.9** | **4.8** | **2.5** | **1.4** | **5.2** | **3.9** | **1.9** | **2.3** | **1.6** | **1.6** |

Table 2: Comparison of methods on layout accuracy evaluation, including position, size, and angle error between generated objects and ground truth (*lower is better*). The unit for position, size and angle is cm, cm, and degree, respectively.

| Method | Bedroom | | Living room | | Dining room | | All | |
|---|---|---|---|---|---|---|---|---|
| | CD | F-score | CD | F-score | CD | F-score | CD | F-score |
| CommonScenes | 0.180 | 0.135 | 0.104 | 0.274 | 0.136 | 0.199 | 0.145 | 0.238 |
| Baseline 1 | 0.091 | 0.443 | **0.077** | 0.503 | 0.076 | 0.506 | 0.080 | 0.485 |
| Baseline 2 | 0.0547 | 0.582 | 0.094 | 0.497 | 0.086 | 0.493 | 0.079 | 0.540 |
| *Incremental3D* | **0.0502** | **0.615** | 0.080 | **0.509** | **0.075** | **0.507** | **0.068** | **0.549** |

Table 3: Comparison of methods on shape accuracy evaluation using Chamfer Distance (*lower is better*) and F-score (*higher is better*).

## 5.2 Layout accuracy

Table 2 summarizes the layout accuracy of different methods. *CommonScenes* shows the lowest accuracy, with average errors of 37.16 cm in position, 43.4 cm in size, and 13.88° in orientation. Baseline 1 achieves errors of 9.45 cm (position), 5.95 cm (size), and 4.55° (orientation). Baseline 2 reports 9.93 cm, 6.63 cm, and 3.88°, respectively. *Incremental3D* attains the highest accuracy, with average errors of only 3.45 cm in position, 2.40 cm in size, and 1.70° in orientation.

Figure 4 provides qualitative layout comparison results. For *CommonScenes*, for example, the chair highlighted in yellow in scene 1 is placed on the right side of the table, whereas the chairs in the ground truth are located on the left. In the third scene, the nightstand (yellow rectangle) collides with the bed in baseline 1 and baseline 2. In contrast, the layouts produced by *Incremental3D* are closest to the ground truth.

Although CommonScenes generates reasonable layouts overall, it exhibits systematic deviations from the ground truth in size, position, and orientation. Its VAE is trained by maximizing the evidence lower bound (ELBO), which encourages distributionally plausible generations rather than precise reconstruction of a specific scene. Baseline 1 and Baseline 2 aim to reconstruct original environments from scene graphs and achieve higher layout accuracy. However, Baseline 1 generates new objects only from local information: despite recomputing node features for both new and existing nodes at each insertion, shallow GNNs are inherently limited in capturing information from indirectly connected nodes (Di Giovanni et al., 2024; Giraldo et al., 2023). For instance, a 3-layer GNN can propagate information at most within a 3-hop neighborhood. Such

one-shot methods are difficult to obtain the global scene embedding, especially in large-scale environments. Baseline 2 restricts new nodes to communicate only with directly connected neighbors, and again generates objects solely from local context. In contrast, *Incremental3D* conditions generation on global scene information through the CLS node. This global embedding aggregates cues such as room symmetry, negative space, collision constraints, and style, enabling more accurate and coherent 3D layout generation compared to the baselines.

## 5.3  Shape accuracy

Table 3 reports the quantitative comparison results for shape accuracy. On average, shapes generated by *CommonScenes* show the poorest global and local geometric alignment with the ground truth, exhibiting the largest mean Chamfer Distance (0.123 m) and the lowest mean F-score (0.21). Baseline 1 achieves a Chamfer Distance of 0.081 m and an F-score of 0.48 across four scenarios, while baseline 2 reports 0.078 m and 0.52, respectively. *Incremental3D* achieves the best global and local geometry alignment with the ground truth, reaching a Chamfer Distance of 0.068 m and an F-score of 0.55.

Figure 4 provides qualitative comparisons of object shapes. In Scene 2, *CommonScenes* produces a noticeably different bed compared to the ground truth. For baseline 1, the table in scene 1 and the wardrobe in scene 3 are clearly misaligned with the reference meshes. Similarly, for baseline 2, the bed in scene 2 does not match the ground truth. In contrast, *Incremental3D*  generates object meshes that are closest to the ground truth.

*CommonScenes* generates plausible and diverse shapes by explicitly modeling a distribution of object geometries conditioned on the scene graph. Since shapes are sampled through a diffusion process in a VQ-VAE latent space, different random seeds or sampling trajectories yield diverse yet valid meshes rather than an exact ground-truth replica. Baseline 1 and baseline 2 achieve better geometric alignment than *CommonScenes*, but the shapes of newly inserted objects rely solely on local information (see layout accuracy section). By comparison, *Incremental3D* leverages the CLS node to inject room-level priors into each new object's conditional distribution, thereby reducing uncertainty in shape prediction and avoiding information decay during deep propagation. As a result, *Incremental3D* converges more effectively to shapes that align with the ground truth.

| Category | Running time | Position | Size | Angle | CD | F-score |
|---|---|---|---|---|---|---|
| Bedroom | 0.024 | 4.3 | 4.8 | 3.8 | 0.067 | 0.559 |
| Living room | 0.031 | 7.5 | 5.1 | 5.7 | 0.073 | 0.496 |
| Dining room | 0.032 | 9.4 | 6.2 | 4.4 | 0.071 | 0.507 |
| All | 0.025 | 5.4 | 4.6 | 3.9 | 0.087 | 0.530 |

Table 4: Performance of ablation experiment without curriculum learning across different room categories.

## 5.4  Robustness Evaluation

A series of experiments under noisy 3D scene graph inputs are conducted to evaluate the robustness of the proposed method. The level of perception noise varies with the environment. For example, CLIO (Maggio et al., 2024) reports a node label accuracy of approximately 98% in apartment environments, while performance degrades to around 73% node precision in office environments and 79% in building-scale environments. To systematically evaluate robustness, we inject synthetic noise into the input 3D scene graphs during data loading. Specifically, we apply noise levels of 2%, 5%, 10%, 20%, and 30% across four settings that simulate common perception errors: incorrect object labels, incorrect edges, missing edges, and missing nodes. For more details about the noise injection methods, refer to the appendix A.2. All robustness evaluations are conducted on the *all* split of the incremental 3D–FRONT dataset, which includes a mixture of living rooms, bedrooms, and dining rooms.

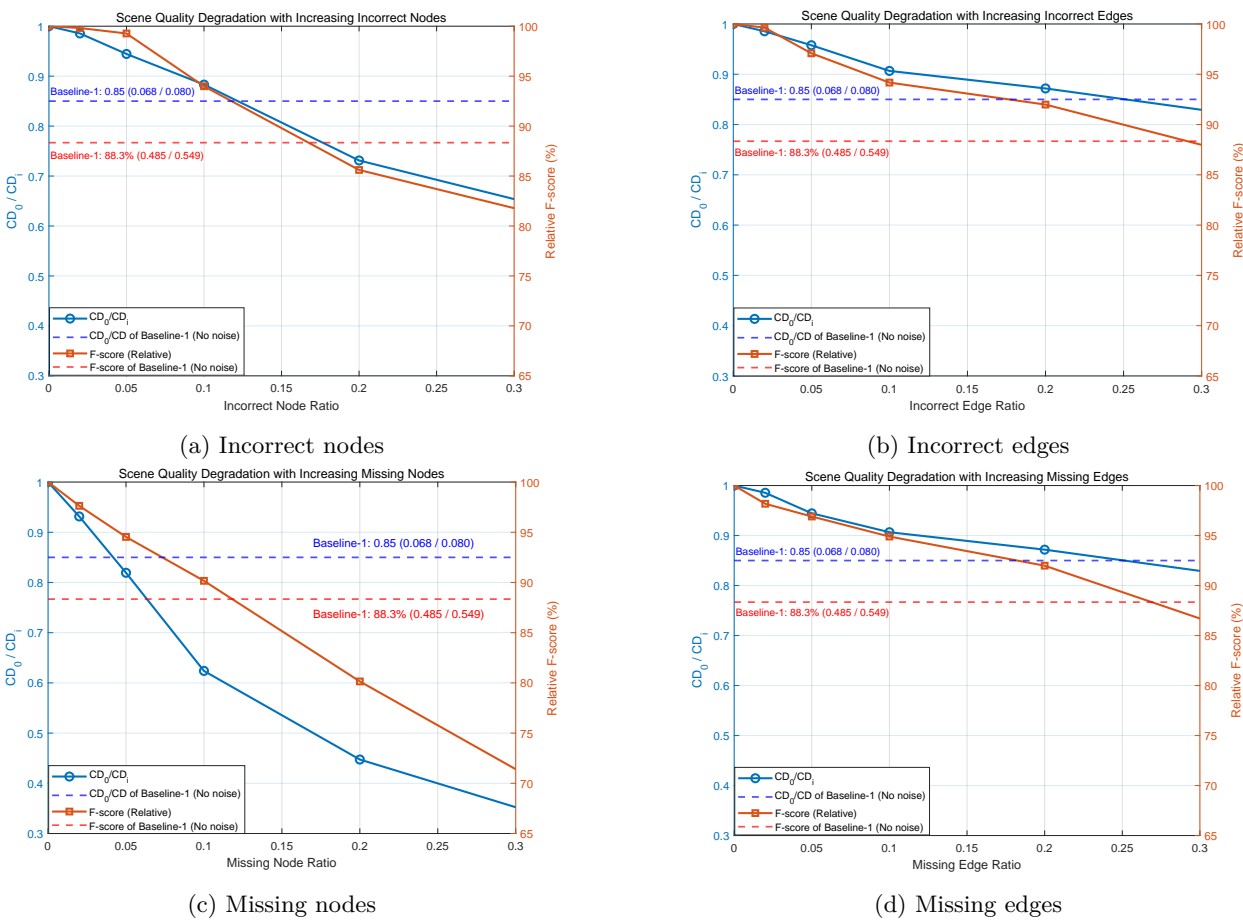

(a) Incorrect nodes

(b) Incorrect edges

(c) Missing nodes

(d) Missing edges

Figure 5: Robustness evaluation under 4 different types of scene graph noise. The horizontal axis denotes the noise level, and the vertical axis represents the global and local geometric similarity between the generated objects and the ground-truth meshes. Incorrect or missing edges have a minor impact on overall scene quality, whereas incorrect node labels lead to a more noticeable degradation. Missing nodes cause more severe performance drops. Under low-noise conditions, *Incremental3D* achieves scene quality comparable to noise-free Baseline–1.

Chamfer Distance (CD) and F-score are used to evaluate global geometry similarity and local geometry similarity between the generated shape and the ground truth shape. The original numerical results for different noise levels are provided in the appendix A.2. A larger CD indicates worse scene quality. To visualize the degradation of global scene consistency as noise increases, we report a normalized metric $CD_0/CD_i$, where $CD_0$ denotes noise-free CD, and $i$ iterates over the 5 noise levels. To measure how local geometry similarity degrades with noise, we use $F\text{-score}_i/F\text{-score}_0$ to illustrate the degradation of perception, where $F\text{-score}_0$ is the noise-free F-score. For missing-node scenarios, the absent objects are represented by a single point when computing the CD against the ground-truth mesh, and the corresponding F-score is set to zero.

Figure 5 summarizes the results under different noise conditions. We also report the performance of Baseline–1 under noise-free inputs for reference. The results show that incorrect or missing edges have only a minor impact on overall scene quality, whereas incorrect node labels lead to more noticeable degradation. For example, with 30% incorrect node labels, the local geometry similarity drops by 18.2%. Missing nodes cause a more severe degradation, indicating substantial loss of structural information; for instance, when 30% of nodes are missing, the local geometry similarity decreases by 28.6%. Nevertheless, under low-noise conditions typical of well-organized indoor environments, the proposed method remains robust. For example, with 10%

incorrect node labels, *Incremental3D* achieves scene quality comparable to Baseline–1 under noise-free inputs. Similarly, when 5% of nodes are missing, the resulting scene quality remains close to that of Baseline–1 with clean scene graphs.

### 5.5 Ablations

Baseline 2 can be regarded as an ablation experiment of CLS node, where the CLS node is removed and new nodes communicate only with their directly connected neighborhood at each incremental step. Results in Table 1, 2, and 3 have demonstrated the importance of the CLS node for incremental scene generation.

To further evaluate the effectiveness of curriculum learning, this ablation experiment retains the CLS node for global context aggregation but is trained without applying the curriculum learning strategy. In this setting, the number of steps is randomized across different batch scenes. Table 4 summarizes its performance in terms of running time and scene accuracy. The average running time across all categories is 0.028 seconds. Compared with *Incremental3D*, layout accuracy degrades on average from (3.45 cm, 2.40 cm, 1.70°) in position, size, and angle errors to (6.65 cm, 5.18 cm, 4.45°), while shape accuracy decreases from (0.068, 0.55) in Chamfer Distance and F-score to (0.0745, 0.52).

## 6 Potential Applications & Limitations

*Incremental3D* opens new possibilities in applications such as telepresence and teleoperation, with scene graphs described in JSON formats and streamed. For example, teleoperation (Tefera et al., 2024; Naceri et al., 2021) techniques rely on continuous image and/or point-clouds streaming from one side to the other, requiring substantial bandwidth and adding significant end-to-end delays, often ranging from hundreds of milliseconds to several seconds, or even minutes (Pop et al., 2022). Such delays make remote control or collaboration difficult. In contrast, a low-latency, low-bandwidth pipeline may be achievable by transmitting text-based, lightweight, incremental 3D scene-graph updates, instead of raw sensory data. The detailed pseudocode for the incremental scene graph construction pipelines are provided in the appendix A.1. The resulting scene-graph summaries can be efficiently transmitted to the operator side over standard 4G or Wi-Fi connections. On the operator side, *Incremental3D* can effectively generate the immersive 3D scene.

While *Incremental3D* demonstrates competitive performance on simulated incremental sequences, its evaluation is limited by the absence of publicly available incremental real-world 3D scene datasets. Our experiments are conducted on synthetic data derived from SG-FRONT (Zhai et al., 2023), which does not fully capture the sensor noise and perception errors typically present in real-world environments. Moreover, in open-world environments, accurate scene graph construction remains an unsolved problem, particularly in the presence of small objects and severe occlusions, which often lead to missing or incorrect object labels. Consequently, *Incremental3D* is primarily intended for well-organized indoor environments at this stage, where object layouts are relatively structured and scene graphs can be constructed with reasonable reliability. Furthermore, *Incremental3D* adopts a database retrieval strategy for object shapes rather than direct shape synthesis, which inevitably limits its open-set generalisation ability.

## 7 Conclusions

In this work, we introduced *Incremental3D*, a framework for real-time 3D scene generation from incrementally evolving scene graphs. Unlike conventional graph-based scene generation approaches that regenerate the entire scene after each update, *Incremental3D* performs localized updates, generating only newly inserted objects, while preserving previously synthesized content. By augmenting the scene graph with a global CLS node, the framework captures a holistic scene representation that conditions the generation of new objects on both local features and global context, enabling consistent and coherent scene updates.

Extensive experiments demonstrate that the proposed approach achieves real-time performance at 38 Hz, while maintaining high layout and geometric accuracy. Quantitative evaluations show that *Incremental3D* improves position, size, and orientation prediction compared with existing baselines, while also producing shapes that better align with ground-truth geometry. These results indicate that combining incremental

generation with global contextual reasoning provides an effective strategy for accurate and scalable 3D scene synthesis from evolving scene graphs.

More broadly, *Incremental3D* highlights the potential of incremental graph-to-3D scene generation as a practical paradigm for real-time environment modelling. By enabling efficient updates from lightweight scene graph representations, the approach might open new possibilities in the future for applications such as teleoperation, telepresence, robotics, and collaborative virtual environments, where responsive and bandwidth-efficient 3D scene reconstruction is essential.

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

# A  Appendix

## A.1  Incremental 3D scene graph construction pipeline

---

**Algorithm 1:** Incremental 3D scene graph construction based on 3D perception

---

**Input**  : RGB-D frames $\{(I_k, D_k)\}_{k=1}^K$, camera poses $\{T_{w \leftarrow c}^k\}_{k=1}^K$
**Output:** Incremental 3D scene graph sequence $\{G_k\}_{k=1}^K$

**1** Initialize global scene graph $G_0 \leftarrow (\mathcal{V}_0, \mathcal{E}_0)$
**2** **for** $k \leftarrow 1$ **to** $K$ **do**
    // 1) Per-frame segmentation
**3**    $\mathcal{S}_k \leftarrow \text{Segment}(I_k, D_k)$
    // 2) Extract per-object features
**4**    **foreach** $s \in \mathcal{S}_k$ **do**
**5**        $F_s \leftarrow \text{ExtractFeat}(s, T_k, D_k)$        // pose, size, color, embedding, etc
    // 3) Search for Neighbor Graph of current frame
**6**    $G_n \leftarrow \text{SearchNeighbors}(T_k)$
    // 4) Predict the scene graph of the current frame using a GNN based on the
        neighbor graph and node feature
**7**    $(\mathcal{V}_k, \mathcal{E}_k) \leftarrow \text{GNN}(G_n, F_s)$
    // 5) Fuse current subgraph into the global scene graph
**8**    $G_k \leftarrow \text{FuseSubgraph}(G_{k-1}, \mathcal{V}_k, \mathcal{E}_k)$
**9** **return** $\{G_k\}_{k=1}^K$

---

---

**Algorithm 2:** Incremental scene graph construction based on 2D perception

---

**Input**  : RGB-D frames $\{(I_k, D_k)\}_{k=1}^K$
**Output:** Incremental scene graph sequence $\{G_k\}_{k=1}^K$

**1** Initialize global scene graph $G_0 \leftarrow (\mathcal{V}_0, \mathcal{E}_0)$
**2** **for** $k \leftarrow 1$ **to** $K$ **do**
    // 1) 2D instance segmentation (e.g., SAM)
**3**    $\mathcal{S}_k \leftarrow \text{Segment}(I_k)$
    // 2) Camera pose estimation
**4**    $P_k \leftarrow \text{EstimatePose}(I_k, D_k)$
    // 3) Per-instance labeling and 3D attribute estimation
**5**    **foreach** $s \in \mathcal{S}_k$ **do**
        // Semantic labeling (e.g., CLIP)
**6**        $\ell_s \leftarrow \text{Classify}(s)$
        // Estimate position $p$, orientation $R$, and size $d$ of objects
**7**        $(p_s, R_s, d_s) \leftarrow \text{Estimate3D}(\ell_s, P_k, I_k, D_k)$
**8**        $v_s \leftarrow \text{CreateNode}(\ell_s, p_s, R_s, d_s)$
**9**        $\mathcal{V}_k \leftarrow \mathcal{V}_{k-1} \cup \{v_s\}$
    // 4) Rule-based relationship inference for the current frame based on layout, etc
**10**    **foreach** $(v_i, v_j) \in \mathcal{V}_k \times \mathcal{V}_k$, $i \neq j$ **do**
**11**        $r_{ij} \leftarrow \text{InferRules}(v_i, v_j)$        // left/right, front/behind, on, near, ...
**12**        **if** $r_{ij} \neq \varnothing$ **then**
**13**            $\mathcal{E}_k \leftarrow \mathcal{E}_k \cup \{(v_i, r_{ij}, v_j)\}$
    // 5) Fuse current-frame subgraph into the global incremental graph
**14**    $G_k \leftarrow \text{FuseSubgraph}(G_{k-1}, \mathcal{V}_k, \mathcal{E}_k)$
**15** **return** $\{G_k\}_{k=1}^K$

---

## A.2 Robustness evaluation

This section of the appendix provides additional details on the noise injection strategies applied to 3D scene graphs, along with the corresponding experimental results for the robustness evaluation.

---

**Algorithm 3:** Injecting Random Incorrect Node Label Noise

**Input** : Correct node labels $\{c_i\}_{i=1}^N$;
noise rate $p_{\text{node}}$
**Output:** Corrupted node labels $\{\tilde{c}_i\}_{i=1}^N$

1 **Init:** $\tilde{c}_i \leftarrow c_i \ \forall i \in \{1, \dots, N\}$

2 **1. Determine number of corrupted nodes**

3 $k \leftarrow \lfloor p_{\text{node}} \cdot N \rceil$

4 **2. Randomly sample nodes to corrupt**

5 Select subset $\mathcal{S} \subseteq \{1, \dots, N\}$ with $|\mathcal{S}| = k$ uniformly

6 **3. Apply random label replacement**

7 **foreach** $i \in \mathcal{S}$ **do**

8 $\quad \lfloor \ \tilde{c}_i \leftarrow \texttt{Sample}(\mathcal{C} \setminus \{c_i\})$

9 **return** $\{\tilde{c}_i\}_{i=1}^N$

---

**Algorithm 4:** Injecting Random Wrong Edge Label Noise

**Input** : Relation triples $\mathcal{T} = \{(s_j, p_j, o_j)\}_{j=1}^E$;
noise rate $p_{\text{edge}}$
**Output:** Corrupted relation triples $\tilde{\mathcal{T}}$

1 **Init:** $\tilde{\mathcal{T}} \leftarrow \mathcal{T}$

2 **1. Determine number of corrupted edges**

3 $k \leftarrow \lfloor p_{\text{edge}} \cdot E \rceil$

4 **2. Randomly sample edges to corrupt**

5 Select subset $\mathcal{E} \subseteq \{1, \dots, E\}$ with $|\mathcal{E}| = k$ uniformly

6 **3. Apply random predicate replacement**

7 **foreach** $j \in \mathcal{E}$ **do**

8 $\quad (s_j, p_j, o_j) \leftarrow \tilde{\mathcal{T}}[j]$

9 $\quad \tilde{p}_j \leftarrow \texttt{Sample}(\mathcal{P} \setminus \{p_j\})$

10 $\quad \tilde{\mathcal{T}}[j] \leftarrow (s_j, \tilde{p}_j, o_j)$

11 **return** $\tilde{\mathcal{T}}$

---

Table 5: Robustness evaluation under different scene graph noise types.

| Noise (%) | Wrong Nodes | | Wrong Edges | | Missing Nodes | | Missing Edges | |
|---|---|---|---|---|---|---|---|---|
| | CD ↓ | F-score ↑ | CD ↓ | F-score ↑ | CD ↓ | F-score ↑ | CD ↓ | F-score ↑ |
| 0 | 0.068 | 0.549 | 0.068 | 0.549 | 0.068 | 0.549 | 0.068 | 0.549 |
| 2 | 0.069 | 0.548 | 0.069 | 0.547 | 0.073 | 0.536 | 0.069 | 0.539 |
| 5 | 0.072 | 0.545 | 0.071 | 0.533 | 0.083 | 0.519 | 0.072 | 0.532 |
| 10 | 0.077 | 0.516 | 0.075 | 0.517 | 0.109 | 0.495 | 0.075 | 0.521 |
| 20 | 0.093 | 0.470 | 0.078 | 0.505 | 0.152 | 0.440 | 0.078 | 0.505 |
| 30 | 0.104 | 0.449 | 0.082 | 0.483 | 0.193 | 0.392 | 0.082 | 0.476 |

We inject controlled noise into the input scene graphs on a per-scene basis during data loading. For each scene, when reading node labels, a predefined proportion of correct object category IDs is randomly replaced with alternative category IDs (see Algorithm 3). When reading relation triples, a predefined proportion of correct predicate IDs is randomly replaced with alternative relation IDs (see Algorithm 4). To simulate missing-edge noise, we randomly select and remove a specified fraction of edges from the incremental scene graph. To simulate missing-node noise, we randomly remove a specified fraction of nodes and delete all edges incident to the removed nodes.

