# OpenReview forum: "Incremental3D: Real-time Incremental 3D Scene Generation with Scene Graphs"
_TMLR — Accepted by TMLR_

### Review · Reviewer_ivud · 2025-12-10

**Summary Of Contributions:**

This manuscript introduces Incremental3D, a framework for incremental graph-to-3D scene generation, motivated by the latency and bandwidth constraints of immersive robotic teleoperation. Unlike existing scene-graph-based 3D generation methods that operate in a single-shot fashion and require regenerating the entire scene upon each graph update, the proposed approach aims to update only newly added objects as the scene graph evolves over time.

The key technical idea is to augment incremental scene graphs with a global context (CLS) node, inspired by the CLS token used in Transformer architectures. This node aggregates global scene information from newly inserted nodes and edges, producing a compact scene-level embedding that conditions the generation of new objects without recomputing unchanged regions. The framework follows an auto-encoder design: a graph-based encoder builds both local node embeddings and a global scene embedding, while a decoder predicts object layout parameters and retrieves corresponding 3D shapes from a database.

To address the lack of real incremental 3D scene datasets, the authors construct synthetic incremental sequences derived from SG-FRONT by clustering objects and determining an insertion order via a Hamiltonian-path approximation, combined with a curriculum learning strategy that gradually increases the number of incremental steps. Experiments compare Incremental3D against representative graph-to-3D generation baselines under metrics of runtime, layout accuracy, and shape fidelity. Results show that Incremental3D achieves real-time performance (around 38 Hz) while improving spatial accuracy over baselines that either regenerate entire scenes or rely purely on local context.

Overall, the work targets an under-explored but practically motivated setting and demonstrates that scene-graph-conditioned 3D generation can be adapted to latency-sensitive, online scenarios.

**Audience:**

Yes

**Audience Explanation:**

Researchers working on graph-based scene generation, robot perception, teleoperation systems, and real-time 3D reconstruction would find the problem setting and findings relevant.

The manuscript introduces an incremental generation paradigm that bridges a practical gap between scene-graph perception and latency-sensitive applications, aligning with TMLR’s interest in practically motivated machine learning.

**Broader Impact Concerns:**

No significant ethical concerns are apparent. Potential misuse is limited and comparable to existing telepresence or simulation technologies. Positive impacts include reduced bandwidth usage and improved safety and usability in remote robotic operation.

**Claims And Evidence:**

Yes

**Claims Explanation:**

The claims regarding real-time incremental generation, reduced latency, and improved layout/shape accuracy over single-shot baselines are supported by quantitative runtime, layout, and shape evaluations, as well as ablation studies on the CLS node and curriculum learning.

However, evidence is restricted to synthetic incremental scenes and does not fully validate performance under real-world perception noise or closed-loop teleoperation settings.

**Requested Changes:**

- Clarify the scope and assumptions under which Incremental3D is expected to operate (e.g., well-organized indoor environments).

- Strengthen discussion on generalization, particularly to noisy or imperfect scene graphs.

- Provide additional justification or analysis of the CLS-based global embedding compared to alternative global aggregation strategies.

- Improve figure clarity and readability in qualitative comparisons.

- Expand limitations to more explicitly address deployment constraints and failure modes.

---

> ### Author Response · Authors · 2026-01-12
> **Response to Reviewer ivud Part 1**
>
> Thank you very much for your valuable feedback on our paper. Your comments have improved both the completeness and the readability of the manuscript. We have uploaded a revised version incorporating your suggestions, with the corresponding changes highlighted in red.
>
> ---
>
> ### Comment 1
> Clarify the scope and assumptions under which Incremental3D is expected to operate (e.g., well-organized indoor environments).
>
> **Answer.**  In open-world environments, accurate 3D scene graph construction remains an unsolved problem: small objects are often missed, and heavily occluded objects are frequently misclassified. These challenges are reduced in well-organized indoor environments, which constitute the primary operating scope of our method. Furthermore, this manuscript focuses on the core incremental scene-generation methodology, albeit aimed at immersive teleoperation application. We have shown the implementation on synthetic data in this manuscript, and real-world testing remains a key part of our future work.
>
> Moreover, to meet real-time requirements, the proposed system generates object geometry via a 3D asset retrieval mechanism. As a result, the shape database is assumed to contain the object categories present in the target scenes, which limits open-set generalization.
>
> In addition, dynamic scenes involving moving or deformable objects are beyond the scope of this work and are not considered in the current formulation.
>
> We have explicitly incorporated these assumptions and limitations into Section~6 of the revised manuscript.
>
> ---
>
> ### Comment 2
> Strengthen discussion on generalization, particularly to noisy or imperfect scene graphs.
>
> **Answer.** Thank you for this suggestion. We have expanded our experimental evaluation to address it. The severity of scene graph noise varies in environments. For example, CLIO [1] reports a node label accuracy of approximately 98\% in real-world apartment environments, while performance degrades to around 73\% in office environments and 79\% in building-scale environments. To systematically assess robustness, we introduce synthetic noise during data loading by perturbing the input 3D scene graphs. Specifically, we evaluate noise levels of 2\%, 5\%, 10\%, 20\%, and 30\% under four representative error types commonly observed in perception systems: (1) incorrect object labels, (2) incorrect object relationships, (3) missing object relationships, and (4) missing object nodes themselves.
>
> The results show that incorrect or missing relationships (edges) have only a minor impact on overall scene quality. Incorrect node labels lead to a more noticeable degradation. For example, with 30\% incorrect node labels, the local geometry similarity drops by 18.2\%. Missing nodes result in the most severe performance drops, as they directly remove objects from the generated scene. Importantly, under low-noise conditions typical of well-organized indoor environments, the proposed method remains robust. For instance, with 10\% incorrect node labels, \textit{Incremental3D} achieves scene quality comparable to Baseline--1 under noise-free inputs. Similarly, when 5\% of nodes are missing, the resulting scene quality remains close to that of Baseline--1 with clean scene graphs. Further details can be found in Section 5.4.
>
> ---
>
> ### Comment 3
> Provide additional justification or analysis of the CLS-based global embedding compared to alternative global aggregation strategies.
>
> **Answer.** Alternative global aggregation strategies include the method of \textit{Max pooling}, which computes $g_{\max}[k] = \max_i \, h_i[k]$, meaning that when generating a new object, only the most dominant nearby feature is considered, while the overall scene context is ignored [2]. Another approach is \textit{Mean pooling}, $g_{\mathrm{mean}} = \frac{1}{N} \sum_{i=1}^{N} h_i$, which assigns equal weights to all objects, without reflecting the fact that closer objects usually have a stronger influence on the newly generated object [3]. Additionally, \textit{Attention pooling} provides a learnable way to extract global information, i.e., $g = \sum_{i=1}^{N} \alpha_i \, h_i$, which can reasonably influence the generation of new objects [4]. However, these pooling-based strategies require storing the features of previously generated objects, making it difficult to scale.
>
> Another class of global information acquisition strategies relies on message passing between the Graph Neural Network (GNN) and all previously generated nodes. However, these methods often suffer from an over-smoothing problem, and shallow GNN layers struggle to capture truly global context[5]. This method has been used as a baseline for comparison.
>
> Compared with these methods, the CLS token provides a simple yet effective solution.
> It does not require storing the shape codes of all previously generated objects, and its computation is lightweight and efficient.
>
> We have included a concise version of this discussion in the manuscript in section 2-C.

---

> ### Author Response · Authors · 2026-01-12
> **Response to Reviewer ivud Part 2**
>
> ### Comment 4
> Improve figure clarity and readability in qualitative comparisons.
>
> **Answer.**  We have  redrawn the figure(s) to improve clarity and readability.
>
> ---
>
> ### Comment 5
> Expand limitations to more explicitly address deployment constraints and failure modes.
>
> **Answer.**
>
> (1) Scene graph noise:
> In cluttered or unstructured environments, existing scene graph construction methods often suffer from erroneous node labels, incorrect relations, as well as missing nodes and edges. The proposed system is therefore designed to operate primarily in well-organized indoor environments, where such perception errors are less frequent. We have expanded the robustness evaluation of our proposed method against different noise levels, showing that performance degrades under severe noise conditions.
>
> (2) Deployment constraints:
> Incremental3D assumes access to object-level scene graphs and a database of 3D assets for shape retrieval. These requirements may not be satisfied in all deployment scenarios, which can limit the applicability of the system in resource-constrained or data-scarce settings, as well as to open-set generalisation. The framework would need to be adapted for deployment with real-world data and hardware setup.
>
> (3) Dynamic scenes or or deformable objects:
> The current system is not designed to handle highly dynamic environments or deformable objects, and thus focuses on static indoor scenes with rigid objects.
>
> We have explicitly incorporated these limitations into the revised manuscript.
>
> ---
>
> ### References:
>
> [1] Dominic Maggio, Yun Chang, Nathan Hughes, Matthew Trang, Dan Griffith, Carlyn Dougherty, Eric Cristofalo, Lukas Schmid, and Luca Carlone (2024). Clio: Real-time task-driven open-set 3d scene graphs. IEEE Robotics and Automation Letters.
>
> [2] Nils Reimers and Iryna Gurevych. Sentence-bert: Sentence embeddings using siamese bert-networks. arXiv preprint arXiv:1908.10084, 2019.
>
> [3] Wu, X., Irie, G., Hiramatsu, K.,  Kashino, K. (2018, October). Weighted generalized mean pooling for deep image retrieval. In 2018 25th IEEE international conference on image processing (ICIP) (pp. 495-499). IEEE.
>
> [4] Hang Zhang, Yeyun Gong, Yelong Shen, Weisheng Li, Jiancheng Lv, Nan Duan, and Weizhu Chen. Poolingformer: Long document modeling with pooling attention. In International Conference on Machine Learning, pp. 12437–12446. PMLR, 2021.
>
> [5] Di Giovanni, F., Rusch, T. K., Bronstein, M. M., Deac, A., Lackenby, M., Mishra, S., \& Veličković, P. (2023). How does over-squashing affect the power of GNNs?. arXiv preprint arXiv:2306.03589.

---

### Review · Reviewer_M2gj · 2025-12-17

**Summary Of Contributions:**

Summary of the paper:

The paper introduces Incremental3D, a framework for generating 3D scenes from scene-graph updates. It's mainly aimed at teleoperation where low latency is key. Instead of regenerating the entire room every time a new object appears, it just updates a global scene representation and generates the new object (predicting its box and shape code). This keeps updates very fast.
Basically, the method adds a CLS node to the graph to aggregate info from new nodes and edges. It uses a Box-GCN and Shape-GCN to predict the layout for new items. They use shape retrieval (via an autoencoder) to keep fidelity high rather than diversity, since it's for teleoperation. They created synthetic sequences from SG-FRONT using clustering and a specific insertion order. They trained it using a curriculum learning schedule. Results show it runs at about 0.026s per update (~38Hz) and has better metrics than the baselines.

Strength:
1.  The motivation is solid. It points out the real problem that exists in robotic teleoperation task and proposes a reasonable yet simple solution: to build the graph incrementally.

2. The runtime is great, hitting ~38Hz on the benchmark.

3. The ablations prove that the CLS node and the curriculum training are actually important for accuracy.

Weakness:
1. The paper operates in a purely synthetic environment. Although it argues this is mainly due to a lack of real-world data, I think the real-world data experiment is critical. First, the real world is much messier than the synthetic, the incremental pipeline might work well on a simple synthetic environment, but it is still highly possible that it doesn't work on the real-world data, due to the accumulated error.  After all, we care about real-world performance, hence the first weakness.
2.  The whole motivation stems from the better teleoperation; however, there is no experiment regarding performing true teleoperation.
3.  The runtime stats aren't fully apples-to-apples since they ran the baseline (CommonScenes) on a different GPU.

**Audience:**

Yes

**Audience Explanation:**

It seems TMLR explicitly does not prefer purely theoretical/methodological work over application papers, as long as the work is within scope.

That said, interest may be concentrated in the subset of TMLR readers who care about 3D generative modeling, graph neural networks, and real-time ML systems; the current experimental setup may feel closer to an engineered module than a broadly general ML contribution.

**Broader Impact Concerns:**

no significant concern

**Claims And Evidence:**

No

**Claims Explanation:**

The evidence holds up for the narrow claim. On their synthetic benchmark, the method achieves fast updates and beats the baselines on layout/shape metrics (Tables 1–3).
However, the broader claims aren't fully supported:
1. They frame this as a solution for real-time teleoperation, but there's no actual teleop study. No robot integration and no evaluation on real-world noise (like missed detections or ID switches). To be fair, the authors do admit this.

2,. The way they built the sequences (clustering + Hamiltonian path) is a reasonable proxy, but it doesn't really match how a robot discovers objects in the wild.

**Requested Changes:**

1.  Strengthen evidence for teleoperation claims: add at least one of
a). a real-robot teleop prototype experiment, or
b). evaluation on real RGB-D scan sequences with automatically produced incremental scene graphs, or
c). a strong simulated pipeline with injected realistic perception noise + ablations on robustness.

---

> ### Author Response · Authors · 2026-01-12
> **Response to Reviewer Part 1**
>
> Thank you for your helpful comments. We appreciate the limitations of our proposed work that you have identified. We hope that our responses to your comments, the dedicated section that we have added to evaluate the model’s robustness against synthetic noise, and the expanded limitations section are satisfactory, and go a fair way in alleviating your concerns.
>
> ---
>
> ### Comment 1
> The paper operates in a purely synthetic environment. Although it argues this is mainly due to a lack of real-world data, I think the real-world data experiment is critical. First, the real world is much messier than the synthetic, the incremental pipeline might work well on a simple synthetic environment, but it is still highly possible that it doesn't work on the real-world data, due to the accumulated error. After all, we care about real-world performance, hence the first weakness.}
>
> **Answer.**   We thank the reviewer for this comment, and acknowledge the concerns raised. The research in the article is indeed limited to synthetic environments. We agree that accurate scene graph generation remains an unsolved problem in real-world environments. Small objects and severe occlusions frequently lead to missing or incorrect semantic labels, which inevitably degrade the quality of generated scenes. However, we would also like to note that the severity of such noise strongly depends on the operating environment. For example, CLIO [1] reports an object label accuracy of up to 98\% in real-world apartment settings.
>
> As stated in the manuscript, the proposed system is primarily designed to operate in well-organized indoor environments. We have extended a discussion on the impact of these limitations, specifically scene graph quality, on the system in the section 6. With this manuscript, we aim to introduce the idea of incremental scene generation for demonstrate that it works well. As noted in Comment \#6 below, we have also expanded our robustness evaluation under different synthetic noise conditions. Nevertheless, the intention with our research is to build on this work, progressing into unstructured real-world environments, going from scene graph construction to incremental scene generation. We would like to position this manuscript as much a standalone contribution, as a first effort in an overall investigation of incremental scene generation for real-world immersive remote teleoperation.
>
> ---
>
> ### Comment 2
> The whole motivation stems from the better teleoperation; however, there is no experiment regarding performing true teleoperation.
>
> **Answer.**  We acknowledge that the current paper does not include real-world teleoperation experiments; this remains an important direction for our future work. To the best of our knowledge, this is a first study in the field of incremental scene-generation framework explicitly aimed at teleoperation. The overall research goal is to explore text-streaming–based immersive telepresence, as opposed to image-streaming-based, allowing scene generation from compact text prompts [2-3]. Our work takes a first step toward this goal, and % by proposing the first scene-generation framework explicitly designed for teleoperation. W
> we plan to extend the proposed framework to real-world teleoperation scenarios in future work.
>
> ---
>
> ### Comment 3
> The runtime stats aren't fully apples-to-apples since they ran the baseline (CommonScenes) on a different GPU.
>
> **Answer.** We agree with the reviewer that the runtime comparison is not strictly apples-to-apples. This discrepancy arises from the intended deployment setting of our method. \textit{Incremental3D} is designed to operate on consumer-grade GPUs and is intended for deployment on a VR headset for real-time robot teleoperation. In contrast, \textit{CommonScenes} cannot be executed on such resource-constrained hardware due to its computational and memory requirements. As a result, we ran CommonScenes on a higher-end GPU, while Incremental3D was on a consumer-grade GPU, testing our proposed method under a more constrained computational setup.

---

> ### Author Response · Authors · 2026-01-12
> **Response to Reviewer  M2gj Part 2**
>
> ### Comment 4
> The broader claims aren't fully supported: They frame this as a solution for real-time teleoperation, but there's no actual teleop study. No robot integration and no evaluation on real-world noise (like missed detections or ID switches). To be fair, the authors do admit this.
>
> **Answer.**  We do acknowledge this limitation and have expanded section 6 to highlight better these constraints. On the point of noise, we take this in conjunction with Comment \#6 below, where we show an expanded robustness evaluation of our proposed method.
>
> ---
>
> ### Comment 5
>  The way they built the sequences (clustering + Hamiltonian path) is a reasonable proxy, but it doesn't really match how a robot discovers objects in the wild.
>
> **Answer.**  We readily acknowledge this limitation. As the reviewer notes, the sequence building and incremental scene graph construction itself is meant to be a proxy to help the research in place of the public datasets and benchmarks that are simply not available in this domain. We agree that a real-world RGB-D camera-based system would function differently, and our ongoing research will expect to show this integration in the near future.
>
> ---
>
> ### Comment 6
> Strengthen evidence for teleoperation claims: add at least one of a). a real-robot teleop prototype experiment, or b). evaluation on real RGB-D scan sequences with automatically produced incremental scene graphs, or c). a strong simulated pipeline with injected realistic perception noise + ablations on robustness.
>
> **Answer.** We appreciate these clear suggestions to improve our manuscript. We have chosen to expand the robustness evaluation of our proposed method, choosing option (c), testing it against controlled synthetic noise injected into the input scene graphs. The severity of scene graph noise varies in environments. For example, CLIO [1] reports a node label accuracy of approximately 98\% in real-world apartment environments, while performance degrades to around 73\% in office environments and 79\% in building-scale environments.
> To systematically assess robustness, we introduce synthetic noise during data loading by perturbing the input 3D scene graphs. Specifically, we evaluate noise levels of 2\%, 5\%, 10\%, 20\%, and 30\% under four representative error types commonly observed in perception systems: (1) incorrect object labels, (2) incorrect object relationships, (3) missing object relationships, and (4) missing object nodes themselves.
>
> The results show that incorrect or missing relationships (edges) have only a minor impact on overall scene quality. Incorrect node labels lead to a more noticeable degradation. For example, with 30\% incorrect node labels, the local geometry similarity drops by 18.2\%. Missing nodes result in the most severe performance drops, as they directly remove objects from the generated scene. Importantly, under low-noise conditions typical of well-organized indoor environments, the proposed method remains robust. For instance, with 10\% incorrect node labels, \textit{Incremental3D} achieves scene quality comparable to Baseline--1 under noise-free inputs. Similarly, when 5\% of nodes are missing, the resulting scene quality remains close to that of Baseline--1 with clean scene graphs. Further details can be found in Section 5.4.
>
> ---
>
> ### References:
>
> [1] Dominic Maggio, Yun Chang, Nathan Hughes, Matthew Trang, Dan Griffith, Carlyn Dougherty, Eric Cristofalo, Lukas Schmid, and Luca Carlone (2024). Clio: Real-time task-driven open-set 3d scene graphs. IEEE Robotics and Automation Letters.
>
> [2] https://openai.com/zh-Hans-CN/index/sora/
>
> [3] Başak Melis Öcal, Maxim Tatarchenko, Sezer Karaoğlu, and Theo Gevers. 2024. SceneTeller: Language-to-3D Scene Generation. In Computer Vision – ECCV 2024: 18th European Conference, Milan, Italy, September 29–October 4, 2024, pp. 362–378.

---

### Review · Reviewer_5zrC · 2025-12-30

**Summary Of Contributions:**

The paper addresses scene graph based teleoperation by proposing a method to strem only the incremental scene graph portion instead of the whole scene graph for low latency operability. Incremental 3D reduces bandwidth usage by orders of magnitude compared to streaming RGB-D video or raw point clouds, making teleoperation possible over poor connections.

The contributions that enable this are:
1. Novel incremental scene graph generation and streaming. To enable this, the paper introduces a global CLS node inspired by the [CLS] token in ViTs into the scene graph to maintain a unified context representation. This global embedding is used to condition each newly added object’s generation and allows for partial scene updates without re-computing the whole scene graph. This is differnt from previous methods.
2. While none-of the architecture is path-breaking, its a well-engineered combination of existing ideas / modules that does yield improved latency.
3. The experiments show in Baseline1 (where the whole scene graph is generated at every step) and Baseline 2 (where the [CLS] token is removed, but the rest of the architecture remains the same) that the Incremental 3D approach does maintain high-quality scene graph structure while allowing it to stream making the re-computing of the scene graph at every step un-necessary. The experiment with Baseline 2 also shows that the [CLS] token is indeed capturing global context and is an important ingredient for scene-graph consistency. This experiment does validate the main thesis of the paper.

**Audience:**

Yes

**Audience Explanation:**

Low-Latency teleoperation by streaming only the updates of scene-graphs seems to be a niche area, so presumably yes the TMLR audience would be interested in this. Especially since the literature on this particular problem seems sparse.

**Broader Impact Concerns:**

1. There are no human studies proposed in the paper, so no human exploitation is involved.
2. From a broader impact perspective, Incremental3D is well-aligned with beneficial applications, particularly in safe and efficient teleoperation. The main concerns are not malicious intent, but over-reliance, hidden uncertainty, and domain mismatch when deployed outside controlled settings. These risks are common to many perception and reconstruction systems and can be mitigated through responsible system integration, transparency, and continued evaluation on real-world data.
3. No privacy concerns exists because the data is synthetic.

**Claims And Evidence:**

Yes

**Claims Explanation:**

There aren't very many papers that address latency issues in tele-operation to the best of my knowledge, but also this is not my field so I might be mistaken. There was a recent one that did [Open-Vocabulary Spatio-Temporal Scene Graph for Robot Perception and Teleoperation Planning](https://arxiv.org/pdf/2509.23107v2), but I'm not sure if it's published somewhere yet. So I believe the paper is addressing a relevant and under-researched problem.

There are approaches that can look at the network layer of this operation and come up with optimizations for this problem, the deep learning approach though is ofcourse also valid and intriguing to me. The main claims in the paper are:

**1. A new problem setting: we introduce incremental graph-to-3D scene generation for teleoperation,
motivated by a low-latency teleoperation system and real-time robot-side incremental scene-graph
perception.**
The paper does introduce a novel graph-to-3D scene generation method that is suitable for low-latency teleoperation. There is one issue in the approach I feel though. Prioritizing autoencoders and shape retrieval favors the high fidelity essential for teleoperation but limits open-set generalization. While this ensures accurate reconstruction of known objects, reliance on a fixed database means the system may fail to represent novel shapes or classes outside the training distribution. Another limitation of the approach seems to be how it handles node-deletion or node-changing. I think the incremental scene graph updates seem to only handle node-addition. In case a node is misclassified, how will it be handled is unclear.

**2. Incremental3D: a real-time framework capable of generating 3D scenes from incremental graph
updates, achieving 38 Hz generation while preserving high spatial and geometric accuracy.**
Incremental 3D does seem to have real-time capability and the low position, size and angle error in Table 2 do seem to suggest that the method is preserving spatial and geometric accuracy.

**3. Introducing a curriculum-learning strategy that progressively increases scene complexity, improving
accuracy on long incremental sequences.**
Curriculum learning does seem to improve learning. Table 4 shows this.

**4. Extensive experiments showing that Incremental3D consistently outperforms single-shot baselines
in layout accuracy, geometric accuracy, and real-time performance.**
Experiments done make sense and are reasonably done. Incremental 3D is compared in three baselines:
1. CommonScenes, standard SOTA single-shot scene generation
2. Baseline 1, which regenerates the full scene at each step (using a  shape retrieval pipeline with some optimizations to make the comparison fair and quick)
3. Baseline 2, an ablation that removes the CLS node  and where new objects use only local neighbor information.
The main limitation in evaluation is the lack of testing on real-world robotic teleoperation data or highly unstructured environments. The experiments are based on the SG-FRONT dataset and an assumed perfect incremental scene graph input. The paper acknowledges the system is expected to work in “well-organized environments”, which suggests it may not yet handle cluttered or highly dynamic scenes. It remains unclear how the approach performs with noisy, partial scene graphs obtained from an actual robot’s perception with missing or misclassified objects. This gap leaves uncertainty about real-world deployment. A stronger evaluation might include running the pipeline with a SLAM + scene graph front-end on real sensor data to confirm robust performance in practice. Also, while the paper’s focus is on the core algorithm, it stops short of demonstrating the approach in a full teleoperation loop. There is no evaluation of how the incremental 3D scene affects a human operator’s performance. This raises the question whether this method can indeed work in real-life scenarios or is just theoretical in nature.

Not sure if this is a novel contribution or has been done before but, the paper provides a method for creating incremental 3D scene datasets by converting static scene graph data into ordered insertion sequences. This is a valuable contribution if no public incremental-scene benchmarks existed. The use of curriculum learning to handle long-horizon sequences is well-motivated and shown to improve training stability. These practical contributions strengthen the paper.

**Requested Changes:**

1. **Pseudocode for explaining the scene graph update**: Maybe add a pseudocode / algorithm outlining the incremental update procedure next to Fig. 2 or as part of it. It is there in the text, but a concise self-contained description of how the system goes from previous state to new state by adding a new object would be helpful and aid understanding.
2. **The Scene Generation process is unclear**: Specifically is it retrieval or generation is opaque to me. Section 3.2 uses a Shape-GCN decoder to output a shape code. I'm unsuse if the shape code is subsequently passed to a learned decoder to produce the mesh, or it retrieves the closest shape from a database. This should definitely be changed to make clear what actually happens.

3. To verify the robustness of the method: Could the authors do an experiment to show how the system handles errors or uncertainty in the input scene graph. In practice, a robot-constructed scene graph may have missing edges or mislabeled nodes. This experiment could be added to the paper. Maybe add noisy edges to the scene graph deliberately and see whether the network is able to correct the introduced errors and whether there is graceful degradation in performance, or performance just plummets on noise addition.

4. On first read, the term “CLS node” might be unfamiliar to some readers. Maybe call it a “global context node” could reinforce its role for people who may not be familiar with the CLS token.

---

> ### Author Response · Authors · 2026-01-12
> **Response to  Reviewer 5zrC Part 1**
>
> Thank you very much for your valuable feedback, in helping improve the completeness and clarity of the manuscript. We have uploaded a revised version that incorporates your suggestions, with the corresponding changes highlighted in red.
>
> ---
>
> ### Comment 1
>  There is one issue in the approach I feel though. Prioritizing autoencoders and shape retrieval favors the high fidelity essential for teleoperation but limits open-set generalization. While this ensures accurate reconstruction of known objects, reliance on a fixed database means the system may fail to represent novel shapes or classes outside the training distribution.
>
> **Answer.** Thank you for this comment, and we agree with the reviewer's perception here. We see this work as a first step in the research towards incremental scene generation for immersive remote teleoperation applications. At this stage, we have focused on well-organised environments, which limits the application of our method and precludes open-set generalisation. Even so, the training dataset we use is based on SG-FRONT, which itself contains more than 5000 scenes, albeit from a structured home environment. We have identified these limitations in the section 6 of the paper. Going forward, as noted by the reviewer as well, given the lack of public incremental-scene benchmarks, we intend to create our own training dataset specifically for this research.
>
> ---
>
> ### Comment 2
> Another limitation of the approach seems to be how it handles node-deletion or node-changing. I think the incremental scene graph updates seem to only handle node-addition. In case a node is misclassified, how will it be handled is unclear.
>
> **Answer.**  We have taken the liberty of coupling this comment with Comment \#6 below, which addresses the suggestion of robustness evaluation of our proposed approach.
>
> ---
>
> ### Comment 3
> The main limitation in evaluation is the lack of testing on real-world robotic teleoperation data or highly unstructured environments. The experiments are based on the SG-FRONT dataset and an assumed perfect incremental scene graph input. The paper acknowledges the system is expected to work in “well-organized environments”, which suggests it may not yet handle cluttered or highly dynamic scenes. It remains unclear how the approach performs with noisy, partial scene graphs obtained from an actual robot’s perception with missing or misclassified objects. This gap leaves uncertainty about real-world deployment. A stronger evaluation might include running the pipeline with a SLAM + scene graph front-end on real sensor data to confirm robust performance in practice. Also, while the paper’s focus is on the core algorithm, it stops short of demonstrating the approach in a full teleoperation loop. There is no evaluation of how the incremental 3D scene affects a human operator’s performance. This raises the question whether this method can indeed work in real-life scenarios or is just theoretical in nature.
>
> **Answer.** We thank the reviewer for highlighting this limitation of our work. Indeed, we acknowledge that these are key assumptions that we make for our research, about well-organized environments and not showing a real-world teleoperation deployment - this is a key component of our ongoing work on this topic. Through this work, our intention is to prove that the concept of incremental scene generation using text-based scene graphs works. We have shown one end of the overall pipeline, i.e., the user-side components, where scene generation happens incrementally based on receiving the incremental scene graphs from the robot-side. As the research progresses, we expect to follow-through with the other components of the overall pipeline, and indeed the evaluation of the system with teleoperation users.
>
> On the point of how the method ``performs with noisy, partial scene graphs obtained from an actual robot’s perception with missing or misclassified objects", we have taken this in conjunction with the Comment \#6 below to expand our robustness evaluation under synthetic noise.
>
> We would request the reviewer to consider this work as much a standalone contribution as a first effort in an overall investigation of incremental scene generation for immersive remote teleoperation.

---

> ### Author Response · Authors · 2026-01-12
> **Response to Reviewer 5zrC Part 2**
>
> ### Comment 4
>  Pseudocode for explaining the scene graph update: Maybe add a pseudocode / algorithm outlining the incremental update procedure next to Fig. 2 or as part of it. It is there in the text, but a concise self-contained description of how the system goes from previous state to new state by adding a new object would be helpful and aid understanding.}
>
> **Answer.**   Incremental scene graph construction methods can be broadly classified into two categories: 3D-perception-based approaches, which infer object semantics from point cloud features, and 2D-perception-based approaches, which recognize object categories from image data. We have added easy-to-follow pseudocode to illustrate the incremental scene graph construction pipeline. Due to space constraints, these two algorithms are provided in the appendix.
>
> ---
>
> ### Comment 5
> The Scene Generation process is unclear: Specifically is it retrieval or generation is opaque to me. Section 3.2 uses a Shape-GCN decoder to output a shape code. I'm unsure if the shape code is subsequently passed to a learned decoder to produce the mesh, or it retrieves the closest shape from a database. This should definitely be changed to make clear what actually happens.}
>
> **Answer.**  We have added the description in Section 3.2 explaining how the shape code is transformed into the final object mesh through a retrieval-based process. In addition, we have revised the system overview (Figure 2) by explicitly including the shape database, making the pipeline easier to understand.
>
> ---
>
> ### Comment 6
>  To verify the robustness of the method: Could the authors do an experiment to show how the system handles errors or uncertainty in the input scene graph. In practice, a robot-constructed scene graph may have missing edges or mislabeled nodes. This experiment could be added to the paper. Maybe add noisy edges to the scene graph deliberately and see whether the network is able to correct the introduced errors and whether there is graceful degradation in performance, or performance just plummets on noise addition.}
>
> **Answer.**  Thank you for this suggestion. We have expanded our experimental evaluation to address it. We inject four types of synthetic noise—(1) incorrect object labels, (2) incorrect edges, (3) missing edges, and (4) missing nodes—into the scene graphs during data loading and then evaluate the resulting scene quality. The noise level varies across environments: CLIO [1] reports a node label accuracy of approximately 98\% in apartment environments, which degrades to around 73\% in office environments and 79\% in building-scale settings. To systematically assess robustness, we apply noise levels of 2\%, 5\%, 10\%, 20\%, and 30\% across the four noise types, simulating common (and progressively severe) perception errors. The results show that incorrect or missing edges have only a minor impact on overall scene quality, whereas incorrect node labels lead to more noticeable degradation; for example, with 30\% incorrect node labels, the local geometry similarity drops by 18.2\%. Missing nodes cause even more severe performance drops. Nevertheless, under low-noise conditions typical of well-organised indoor environments, the proposed method remains robust. For instance, with 10\% incorrect node labels, \textit{Incremental3D} achieves scene quality comparable to Baseline--1 under noise-free inputs. Similarly, when 5\% of nodes are missing, the resulting scene quality remains close to that of Baseline--1 with clean scene graphs. More details can be found in Section~5.4.
>
> ---
>
> ### Comment 7
> On first read, the term “CLS node” might be unfamiliar to some readers. Maybe call it a “global context node” could reinforce its role for people who may not be familiar with the CLS token.}
>
> **Answer.**   In the initial parts of the text (abstract and introduction), we have replaced the term CLS node with \textit{global context node} to avoid confusion. The term CLS node is used only after introducing the CLS-token–related literature.

---

> ### Author Response · Authors · 2026-01-12
> **Response to Reviewer 5zrC Part 3**
>
> ### Comment 8
> Not sure if this is a novel contribution or has been done before but, the paper provides a method for creating incremental 3D scene datasets by converting static scene graph data into ordered insertion sequences. This is a valuable contribution if no public incremental-scene benchmarks existed. The use of curriculum learning to handle long-horizon sequences is well-motivated and shown to improve training stability. These practical contributions strengthen the paper.}
>
> **Answer.** Thank you for this comment. Indeed, to the best of our knowledge, this is a first attempt for incremental scenes in teleoperation. While [4] also applies scene graphs to teleoperation, it deploys them on the remote robot side for autonomous task planning and remains an image-streaming–based teleoperation system. In contrast, our overall research effort in this topic seeks to explore a fundamentally different teleoperation paradigm based on text (scene graph) streaming, aiming to generate scenes from compact text prompts [2-3]. % , inspired by recent advances in generative AI showing that realistic 2D and 3D scenes can be generated from compact text prompts [2-3].
> Our work takes a first investigative step towards this vision by proposing the first scene-generation framework specifically designed for teleoperation.
>
> ---
>
> ### Reference
>
> [1] Dominic Maggio, Yun Chang, Nathan Hughes, Matthew Trang, Dan Griffith, Carlyn Dougherty, Eric Cristofalo, Lukas Schmid, and Luca Carlone (2024). Clio: Real-time task-driven open-set 3d scene graphs. IEEE Robotics and Automation Letters.
>
> [2] Yi Wang, Zeyu Xue, Mujie Liu, Tongqin Zhang, Yan Hu, Zhou Zhao, Chenguang Yang, and Zhenyu Lu (2025). Open-Vocabulary Spatio-Temporal Scene Graph for Robot Perception and Teleoperation Planning. arXiv preprint arXiv:2509.23107.
>
> [3] https://openai.com/zh-Hans-CN/index/sora/
>
> [4] Başak Melis Öcal, Maxim Tatarchenko, Sezer Karaoğlu, and Theo Gevers. 2024. SceneTeller: Language-to-3D Scene Generation. In Computer Vision – ECCV 2024: 18th European Conference, Milan, Italy, September 29–October 4, 2024, pp. 362–378.

---

### Decision · Action_Editor_5DRu · 2026-03-05

**Recommendation:** Accept with minor revision

**Additional Comments:**

The initial reviews were consistent in finding support for some of the paper's core claims (efficiency, reduced latency, and the benefits of the CLS node and curriculum learning) as well as in the lack of evidence to support the implied practical relevance of the work. The authors provided a detailed response to the initial reviews, acknowledging the reviewers' concerns, while adding new experimental results.

Following the discussion period, one reviewer recommended acceptance, another leaned towards acceptance, while the third leaned towards rejection. All three noted the paper's value in studying an under-represented problem and in the demonstrated advantages of the CLS node and curriculum learning. Two of the reviewers felt that there were still issues with how the paper positions the work in the context of robot teleoperation in real-world environments. The AE agrees and feels that it is important that this be resolved prior to publication.

**Audience:**

Yes

**Audience Explanation:**

There is consensus among the reviewers that the paper addresses an important and relatively under-studied problem. As they point out, the introduction of a framework that is capable of low-latency, real-time scene generation will be of interest to researchers working on robot mapping/perception and scene understanding/generation.

**Claims And Evidence:**

No

**Claims Explanation:**

**Claims or suggestions that the proposed method (Incremental3D) provides a solution for real-time robot teleoperation in real-world environments are not supported by the evidence provided. Per the discussions with the authors and reviewers, these claims must be substantiated or removed in a future revision. The other core claims of the paper meet the evidence criterion.**

The paper proposes Incremental3D, a framework that generates 3D scene models from scene-graph updates. Motivated by the problem of real-time teleoperation, Incremental3D performs graph-to-3D scene generation in an incremental fashion each time a new object appears, resulting in greater computational efficiency and reduced latency compared to approaches that regenerate the entire scene to integrate each new observation. More specifically, Incremental3D augments the scene graph with a global classification (CLS) node that aggregates global scene information from new nodes and edges, creating a compact scene-level embedding that conditions the generation of new objects without the need to recompute regions that have not changed. Without access to datasets of real incremental scene generation, the paper constructs synthetic incremental sequences, and trains the model using a curriculum learning strategy that increases sequence length and difficulty during training. The paper compares Incremental3D to different graph-to-3D generation methods and provides ablations to better understand the contributions of the key model components.

The reviewers agree that the paper provides adequate support for several of its key claims. For one, the experimental results demonstrate that Incremental3D is capable of generating 3D scene graphs in real time, while preserving accuracy. Second, the experiments validate the benefits of the proposed curriculum learning strategy with regard to improvements in accuracy and increased scene complexity. Third, the ablation study demonstrates the importance of the global classification (CLS) node.

At the same time, there was consensus among the reviewers that the suggested capabilities of Incremental3D when used in practice are not supported. While the work is heavily motivated by the problem of robot teleoperation, the paper does not include any real-world robot teleoperation experiments. Further, it assumes that the environments are well organized, which is typically not the case in practice as most environments are highly cluttered and dynamic. The authors acknowledged these issues during the discussion phase. They performed additional experiments that evaluate the method's robustness to different types of noise (i.e, incorrect object labels, incorrect edges, missing edges, and missing nodes), and otherwise comment that the focus of the paper is to conceptually show the potential of incremental scene generation using text-based scene graphs.

The AE sees the conceptual value of the work, but agrees that it is inappropriate to suggest that Incremental3D provides a solution for real-time robot teleoperation without teleoperation results, integration on a robot, or experiments on real-world environments and real-world noise.

---

> ### Author Response · Authors · 2026-03-16
> **A revised version with a refined scope, shifting the focus from teleoperation to 3D scene generation based on incremental scene graphs.**
>
> Dear AE and Reviewers,
>
> Thank you for the positive news that our article has progressed towards acceptance with minor edits. We appreciate the constructive feedback and helpful suggestions in the letter, and we have carefully revised the manuscript to address the concerns.
>
> To address the main concern, we have narrowed the scope of the paper and disconnected it from the robotic teleoperation application. Instead, the revised manuscript re-focuses on the core problem studied in this work, that of 3D scene generation from incremental scene graphs, which remains an underexplored problem.
>
> We clarify that existing graph-based 3D scene generation methods are fundamentally single-shot, requiring regeneration of the entire scene when a new object is inserted, which leads to significant computational overhead and latency. Our approach, Incremental3D, augments the scene graph with a global CLS node to capture holistic scene context and enables incremental generation of newly inserted objects without recomputing unchanged regions. In addition, the model is trained using a curriculum learning strategy based on the incremental insertion steps to further improve performance. The effectiveness of the proposed framework and the CLS node is demonstrated in Tables 1–3 and Figure 4 in the experimental section. Table 4 further highlights the benefits of the curriculum learning strategy.
>
> We hope that the revisions undertaken will satisfactorily address the requested edits for the manuscript. As advised in the decision email from TMLR Editors-in-chief, we are now submitting the deanonymized camera ready revision of the paper. Kindly let us know if any further clarifications or final revisions are needed for the manuscript.
>
> Thank you very much again for your time and consideration.
>
>
> Best regards,
>
> The Authors